# TRIM25: A Global Player of Cell Death Pathways and Promising Target of Tumor-Sensitizing Therapies

**DOI:** 10.3390/cells14020065

**Published:** 2025-01-07

**Authors:** Wolfgang Eberhardt, Usman Nasrullah, Josef Pfeilschifter

**Affiliations:** Institute of General Pharmacology and Toxicology, Goethe University Frankfurt, 60590 Frankfurt, Germany; nasrullah@em.uni-frankfurt.de (U.N.); pfeilschifter@em.uni-frankfurt.de (J.P.)

**Keywords:** cell death pathways, therapy resistance, TRIM25, tumorigenesis

## Abstract

Therapy resistance still constitutes a common hurdle in the treatment of many human cancers and is a major reason for treatment failure and patient relapse, concomitantly with a dismal prognosis. In addition to “intrinsic resistance”, e.g., acquired by random mutations, cancer cells typically escape from certain treatments (“acquired resistance”) by a large variety of means, including suppression of apoptosis and other cell death pathways via upregulation of anti-apoptotic factors or through inhibition of tumor-suppressive proteins. Therefore, ideally, the tumor-cell-restricted induction of apoptosis is still considered a promising avenue for the development of novel, tumor (re)sensitizing therapies. A growing body of evidence has highlighted the multifaceted role of tripartite motif 25 (TRIM25) in controlling different aspects of tumorigenesis, including chemotherapeutic drug resistance. Accordingly, overexpression of TRIM25 is observed in many tumors and frequently correlates with a poor patient survival. In addition to its originally described function in antiviral innate immune response, TRIM25 can play critical yet context-dependent roles in apoptotic- and non-apoptotic-regulated cell death pathways, including pyroposis, necroptosis, ferroptosis, and autophagy. The review summarizes current knowledge of molecular mechanisms by which TRIM25 can interfere with different cell death modalities and thereby affect the success of currently used chemotherapeutics. A better understanding of the complex repertoire of cell death modulatory effects by TRIM25 is an essential prerequisite for validating TRIM25 as a potential target for future anticancer therapy to surmount the high failure rate of currently used chemotherapies.

## 1. Introduction

Evasion of apoptosis and the activation of cell survival programs are common characteristic features of cancer cells, regardless of the cause and types [1,2]. In addition, impaired cell death mechanisms can render tumor cells more resistant towards toxic insults, and thus, they represent a main reason for anticancer drug resistance [3]. Impaired apoptotic control allows survival of tumor cells even with otherwise lethal mutations and defects in the DNA damage response which commonly increase the risk of selection of highly aggressive tumor cells with strong tumorigenic properties, e.g., those cells acquired with an increased invasive potential (for a review, see [4,5]). Consequently, the restoration of drug-induced cell death mechanisms became one of the earliest concepts and realistic goals of novel cancer therapeutics. In addition to apoptosis, the first regulated form of cell death described in the 1970s, further common modes of regulated cell death forms, including necroptosis, pyroptosis, ferroptosis, and autophagy, have been discovered and underlying signal pathways have been deciphered at the molecular level (for a review, see [6,7]). Dysregulations of these alternative cell death modalities are likely to be involved in the etiology of chemotherapeutic-resistant cancers as well.

Mechanistically, besides mutations in the pleiotropic tumor suppressor p53 observed in approximately 50% of solid tumors, the evasion from apoptosis is mainly attributed to either gain-of-function mutations or overexpression of anti-apoptotic effector proteins, including members of the B cell CLL/lymphoma 2 (Bcl-2) protein family as Bcl-2, Bcl-xl, and Mcl-1, inhibitors of apoptotic proteins (IAPs) such as survivin, which all have an ultimate impact on drug resistance (for a review, see [5,8]). Consequently, targeted inhibition of these factors seems an attractive avenue for novel antitumor therapies with chemosensitizing effects. Thanks to the enormous progress in understanding the molecular mechanisms utilized by tumor cells to escape from apoptotic cell death, the implementation of novel therapeutics to the clinic has been successfully realized and is exemplified by the small molecule Bcl-2 inhibitor Venetoclax approved for the treatment of chronic lymphatic leukemia. The pro-survival protein Bcl-2 is overexpressed in more than half of all human cancers, regardless of type, and therefore, besides the tumor suppressor gene p53, it is one of best-characterized apoptosis regulators [9]. Nonetheless, therapy resistance and cancer reoccurrence are still major obstacles for the long-term benefits of almost every clinically used antitumor therapeutic. Unfortunately, most novel cell death/cell survival modulatory approaches have failed to pass complete clinical phases due to severe side effects [5]. Unwanted side effects of a drug are in most cases multifactorial, ranging from the genetic background of tumor cells to diverse transiently acquired mechanisms (for a review, see [10]). Therefore, a more promising approach to interfere with drug resistance may utilize the interference with anti-apoptotic cell responses induced by chemotherapeutic drugs while leaving constitutively active cell death pathways untouched [10]. Conceptually, approaches to reach these requirements may, therefore, better target therapy-induced events, such as posttranslational modifications (PTMs), by interfering with the enzymes that carry out respective modifications. Evidence has accumulated demonstrating that aberrant turnover of regulatory proteins implied in the control of, e.g., DNA repair, apoptosis, morphogenesis, transcriptional regulation, protein quality, and others due to a deregulated ubiquitin proteasome system (UPS), is crucially involved in the etiology of carcinogenesis (for a review, see [11]). The UPS summarizes an enzymatic cascade which is executed by three different types of enzymes, including E1 (ubiquitin activating enzymes), E2 (conjugating enzymes), and E3 ubiquitin ligases. Currently, only 2 different E1 enzymes are known, while more than 40 E2 enzymes have been characterized. Interestingly, an estimated 500–1000 E3 ligases should be present in the human genome [12,13]. A large number of different UPS enzymes allows a multitude of combinations, and therefore, a high functional diversity. The specificity of different E3 ligases is achieved by diverse mechanisms, with most of them including the recognition of short amino acid sequnences or chemically modified substrates summarized as substrate degradation signals (degrons) [14]. Depending on the kind of ubiquitination, different intracellular processes are exhibited by E3 ligase enzymes. In addition to K48- and K11-linked polyubiquitination, the most common ubiquitin modifications, which confer protein degradation by the 26S proteasome [15], tagging of substrates, e.g., via K63-linked ubiquitin chains, can control many non-degradative processes, e.g., receptor internalization, intracellular signaling pathways such as NF-κB, DNA repair, immunity, and cell survival [11,16]. In contrast, the detailed functions of other, less frequently observed ubiquitination linkages (K6, K27, K29, and K33) are only marginally known. Since ubiquitination modifications affect an enormous repertoire of cellular processes; thus, it is not surprising that dysregulation of these PTMs can have detrimental consequences and is, therefore, frequently linked with a large variety of human disorders, including inflammation and cancer [17].

One potential class of enzymes which has drawn increased attention during the last decade is the tripartite motif (TRIM) family of E3 ligases. Among the RING type of E3 ligases, TRIMs represent the largest and best-characterized subfamily, with more than 80 TRIM proteins currently known to be present in humans [11,18]. TRIM proteins are structurally characterized by different clusters of domains, including an N-terminal RING finger domain (with the exception of TRIM16 and TRIM20, which lack a characteristic RING domain) relevant for the catalytic function of the enzymes, one to two zinc finger motifs, summarized as B-box domains, and a coiled-coil (CC) domain. These kinds of domains mediate protein–protein interactions and hetero- or homodimerization, respectively (Figure 1). At the C-terminus, most TRIM proteins contain one or more variable domains, with the SPRY domain representing the most common variant (Figure 1) [19,20]. This domain is also involved in protein–protein interaction, though only in some TRIM members (TRIM25, TRIM56, and TRIM71), additionally relevant for RNA binding as well [21]. Depending on the composition and arrangement of the different domains, TRIM proteins are further divided into 11 subfamilies [22]. Functionally, due to their role in substrate ubiquitination, TRIMs are pivotally involved in the control of cellular key functions, e.g., transcription, cell growth and differentiation, migration, cell death, and immune regulation [11,23]. Consequently, alterations in TRIM expression are implicated in the pathogenesis of many human diseases, including cancer [18,24]. Some individual TRIM members are not only relevant for tagging of proteins by ubiquitinylation, but additionally implied in the SUMOylation and NEDDylation [11,25].

In the last decade, among the large number of TRIM family members, TRIM25 deserved increased attention for several reasons. Mainly, TRIM25 has antiviral capacities, which are mainly achieved by the induction of canonical retinoic acid inducible gene 1 (RIG-1)/type I interferon signaling, and are relevant for defense against different RNA viruses, such as Dengue virus [26], respiratory syndrome virus (PRRSV), human papillomavirus (HPV), and influenza A virus [27,28]. This underlines the high immunoregulatory function of this particular TRIM member. Interestingly, TRIM25 itself is target of severe acute respiratory syndrome Coronavirus-2 (SARS-CoV-2), mainly through the inhibition of TRIM25–RIG-1 interaction [29]. The binding affinity to viral RNA by TRIM25, which has been expanded to mammalian RNAs, is one of the common features of the newly defined TRIM class, with an additional RNA binding capacity (among them TRIM25, TRIM56, and TRIM71) [21]. Additionally, TRIM25 plays pivotal roles in different aspects of tumorigenesis, and data from several laboratories have demonstrated that dysregulated TRIM25 participates in the development of various human cancers (for a review, see [30,31,32]). Accordingly, TRIM25 is overexpressed in many human cancers and a high tissue expression level of TRIM25 often correlates with an unfavorable prognosis and outcome of patients, thus highlighting the potential of TRIM25 as a biomarker and a valid therapeutic target for novel anticancer therapies.

Given that TRIM25 in most cases propagates chemotherapy resistance particularly through the inhibition of apoptosis, this review will summarize up-to-date knowledge of the heterogeneous mechanisms by which TRIM25 interferes with different cell death modalities, including apoptosis, pyroptosis, necroptosis, ferroptosis, and autophagy.

## 2. Regulation of Major Cell Death Pathways by TRIM25

### 2.1. Regulation of Apoptosis by TRIM25

Apoptosis, the earliest programmed cell death process described in the literature, is morphologically characterized by cell shrinkage, extensive blebbing of plasma membranes, chromatin condensation, and DNA fragmentation in the nucleus [33]. In clear contrast to necrosis, apoptosis is an immunological silent process without induction of inflammation. Mechanistically, apoptosis is triggered either by extracellular signals or intracellular stimuli, and accordingly, executed by the extrinsic pathway, also referred to as the death receptor (DR)-mediated pathway, and the intrinsic, mitochondrial pathway, respectively. DRs such as Fas, TNF receptor 1, and TRAIL receptor, upon binding to their respective ligands (Fas ligand, TNF, and TRAIL), initiate the extrinsic pathway by direct recruitment of caspase activation platforms (for a review, see [34]). In contrast, the intrinsic pathway of apoptosis is initiated by the loss of integrity of the outer mitochondrial membrane, leading to the release of proapoptotic factors such as cytochrome c and second mitochondria-derived activator of caspase (SMAC) into the cytoplasm. Cytochrome c, together with the apoptosis protease activating factor 1 (Apaf-1), induces the assembly of the apoptosome complex followed by the activation of pro-caspase 9. Importantly, both pathways converge at the activation of executioner caspases, which act downstream of initiator caspases, ultimately leading to the degradation of intracellular target proteins and destruction of the cell (for a review, see [35]).

#### 2.1.1. TRIM25 and p53-Dependent Survival Pathways

The tumor suppressor protein p53 is a critical regulator of the cell cycle, maintenance of genome stability, and apoptosis, making it additionally a key player in chemotherapy resistance [36]. Accordingly, p53 mutations account for up to 50% of all human cancers, and in another additional 40% of tumor patients, upstream regulatory proteins of p53 are impaired [37]. Meanwhile, alternately to the well-described ubiquitin E3 ligase murine double minute 2 (Mdm2), several TRIM proteins are critical regulators of p53, most of them having a negative impact on p53, as many target the p53 protein for proteasomal degradation [10]. Thus, p53-inhibitory TRIM members, including TRIM23 [38,39,40], TRIM24 [41,42,43], TRIM25 [44,45], TRIM28 [46,47], TRIM29 [48,49], and TRIM59 [50,51], are frequently overexpressed in human cancers. TRIM25, which is transcriptionally induced by estrogens and plays critical roles in the promotion of hormone-dependent tumors, in particular those of the breast, the prostate, and the endometrium, is synonymously called “Estrogen-responsive finger protein (Efp)”.

Paradoxically, in the colorectal cancer cell line HCT116, TRIM25 exerts a dual function in the p53/Mdm2 circuit via two apparently opposing mechanisms [52]. On the one hand, it impairs the formation of a ternary complex of Mdm2, p300, and p53, which is relevant for p53 ubiquitination and subsequent proteasomal degradation, and thereby, TRIM25 can stabilize the p53 protein. On the other hand, the same complex is required for p53 acetylation, which is a critical posttranslational modification for enhanced transcriptional activity by p53, mainly through the recruitment of transcriptional coactivators, e.g., p300, to the promoters of p53 responsive genes [52]. In this case, the inhibition of apoptotic cell signaling through a blockade of p53-dependent transcription dominates for the final impact of TRIM25. Nevertheless, inhibition of p53 degradation by Mdm2 is considered a valid therapeutic strategy to regain high p53 levels in tumor cells. This can principally be achieved either by direct inhibition of Mdm2 ubiquitin ligase activity or by preventing the physical interaction of both proteins [10]. In contrast to p53 stabilization by TRIM25 described in HCT116 cells, complexes between TRIM25, Mdm2, and p53, observed in human lung cancer tissue and lung cancer cells which are characterized by low p53 abundance, indicate that in these cases TRIM25 does rather promote Mdm2-mediated proteasomal degradation of p53 [44]. Consequently, depletion of TRIM25 resulted in a significant increase in apoptosis sensitivity to the topoisomerase inhibitor doxorubicin. In addition to elevating apoptosis sensitivity of lung cancer cells, the knockdown of TRIM25 markedly reduced other tumorigenic parameters, including proliferation and migration [44].

Another study employed a comparative proteomic strategy to screen for ubiquitination-associated factors potentially involved in cisplatin resistance in human lung adenocarcinoma cells, and identified TRIM25 as a candidate protein displaying significant overexpression in cisplatin resistant A549 cells (A549/CDDP) when compared with parental A549 cells (Figure 2). Accordingly, RNA-interference-mediated knockdown of TRIM25 restored the sensitivity towards cisplatin-induced apoptosis in the cisplatin-resistant A549/CDDP cells [53]. Mechanistically, TRIM25-associated drug resistance was linked with downregulation of the scaffold protein 14-3-3σ (Figure 2), which itself is a known target of TRIM25-dependent proteasomal degradation in breast cancer by depending on the ubiquitin conjugating enzyme UbcH8 [54]. Consistently, downregulation of this chaperone has been observed in various carcinomas. In most cases, 14-3-3σ, by accelerating the proteolytic turnover of Mdm2 through enhancing auto-ubiquitination of Mdm2, leads to the stabilization of p53, and subsequently, to the activation of p53-related tumor-suppressive functions (Figure 2) [55,56]. Accordingly, sensitizing effects were determined after cisplatin-induced apoptosis in A549/CDDP cells after TRIM25 knockdown, correlated with low Mdm2, but elevated p53 levels and sensitization were completely impaired after the additional knockdown of 14-3-3σ [53]. Meanwhile, by means of biophysical and bioinformatic analysis, the binding interface of TRIM25 and 14-3-3σ was identified, and accordingly, peptides which match this amino acid sequence may beneficially be used for the design of inhibitors of TRIM25-14-3-3σ interactions as potentially novel class of anticancer drugs [57]. These data further indicate that TRIM25 confers apoptosis resistance indirectly, through interfering with the 14-3-3σ-triggered proteasomal degradation of the p53 inhibitor Mdm2. Since 14-3-3σ is prominently involved in cell cycle regulation and G2 arrest, TRIM25-mediated degradation of this chaperone in addition to chemotherapy resistance seems highly relevant for other oncogenic key functions [54]. In this regard, it is worth mentioning that in the context of breast cancer, the use of TRIM25 targeting DNA-based “chimeric siRNA”, in which sequence fragments are substituted with DNA, is supposed to have fewer off-target effects and less immunogenic potential and efficiently suppressed proliferation and cell cycle progression of MCF-7 cells as well as in vivo growth of MCF-7-derived tumors in nude mice [58]. Again, the beneficial effects on tumor growth by TRIM25 knockdown were supposed to be mainly due to the upregulation of the negative cell cycle regulator 14-3-3σ. This study supports the concept of employing chimeric TRIM25-specific chimeric siRNA as a useful treatment alternative in breast cancer therapy [58].

Alternatively, the direct targeting of the proto-oncogenic E3 ligase Mdm2 is the basis of the anti-infective drug nitroxoline (NXQ), which is potentially useful for therapy of small cell lung cancer [59]. Similarly, in a mouse xenograft model of multiple myeloma, NXQ was shown to be most effective in potentiating the apoptotic effects by doxorubicin and lenalidomide, which are currently used as therapy alternatives for treatment of the disease [60]. Mechanistically, NXQ itself was found to increase the expression of the tumor suppressor p53, mainly through downregulation of TRIM25, which is highly expressed in multiple myeloma [60]. It is, therefore, tempting to speculate that the chemo sensitizing effects observed in small cell lung cancer by this compound are also through an inhibition of TRIM25-mediated 14-3-3σ degradation, leading to an impaired degradation of p53 by Mdm2. Although the molecular mechanisms for downregulation of TRIM25 by NXQ await further investigation, these studies highlight the potential benefit of targeting the TRIM25–14-3-3σ–Mdm2–p53 axis as a strategy for novel antitumor therapies.

In addition to regulating p53 ubiquitination and proteasomal degradation or p53 transcriptional activity via impaired recruitment of transcriptional co-activators, TRIM25 can affect the subcellular translocation of p53, which is relevant for resistance towards docetaxel-induced apoptosis [61]. Notably, the intracellular localization of p53 is mainly determined by its ubiquitination status. Conversely, polyubiquitination by, e.g., the E3 ligase Mdm2, promotes p53 proteasomal degradation, monoubiquitination of p53 can promote its nuclear export [62]. In addition, p53 is a target of SUMOylation by the SUMO E3 ligase RAN-binding protein 2 (RanBP2), which, in complex with the androgen-induced GTPase-activating protein-binding protein 2 (G3BP2), promotes the nuclear export, and consequently, hinders the transactivation capacity of p53 in the nucleus [63]. Importantly, G3BP2 at the same time represses polyubiquitination of p53 by Mdm2. Interestingly, in this scenario, TRIM25 seems to act as a key regulator of the balance between both opposing posttranslational modifications. In a search for G3BP2-interacting proteins in prostate cancer LNCaP cells, RanBP2 and TRIM25 were among the top interacting factors. Mechanistically, TRIM25 acts as a conductor of G3BP2-mediated p53 export by promoting sumoylation of p53 via RanBP2, but exerts a negative influence on p53 ubiquitination [61]. Notably, silencing of TRIM25 did not only diminish the cytoplasmic abundance of G3BP2 but also reduced total G3BP2 levels, suggesting a dual role of TRIM25 in both protein expression and subcellular localization of G3BP2 in prostate cancer cells [61]. Accordingly, the authors could demonstrate that TRIM25 knockdown caused a significant reduction in tumor growth due to increased p53 activity when using a mouse xenograft model of prostate cancer [61]. In summary, overexpression of TRIM25 in prostate cancer promotes cell survival and cell proliferation mainly through the activation of G3BP2/RanBP2-mediated SUMO conjugation required for nuclear p53 export.

Another molecular basis for increased p53 inactivation by TRIM25 to be mentioned is the upregulation of TRIM25 expression by the oncogenic small nuclear RNA (snoRNA) and C/D Box 15B (SNORD15B) [64]. In endometrial cancer, SNORD15B exerts diverse tumorigenic features by enhancing cell proliferation and migration, but also through the inhibition of apoptosis. Generally, SNORDs are located close to introns and are released during the splicing process. In most cases, SNORDs are targets of exonuclease-mediated degradation, but complex formation with specific nucleolar proteins, including the nucleolar protein 5 (NOP58), non-histone chromosomal protein 2-like 1 (NHP2L1), nucleolar protein 56 (NOP5A), and fibrillarin (FBL), can protect SNORDs from exonuclease-mediated degradation [65]. While SNORDs have originally been described to mediate 2′-O-ribose methylation of ribosomal (r)RNAs, recent studies unraveled a much broader repertoire, which may be relevant for tumor development as well.

Pathologically, SNORD15B is significantly upregulated in endometrial cancer and associated with poor patient survival. Accordingly, increased SNORD15B levels are also found in different endometrial cancer cell lines. Originally, SNORD15B could be identified as a TRIM25-bound RNA target by UV crosslinking immunoprecipitation (IP) approaches, but was not deeper analyzed [21]. In addition, co-IP experiments revealed a physical interaction between TRIM25 and p53 in the cytoplasm of endometrial cancer cells, which was further increased in cells overexpressing SNORD15B [64]. Data from this study suggest that SNORD15B interferes with the nuclear translocation of p53 mainly through upregulation of TRIM25, resulting in the inhibition of p53-dependent gene expression. SNORD15 could, therefore, offer another promising target for treatment of endometrial and other cancers. Although snoRNAs are implied in the regulation of genomic stability as well as in mRNA splicing and translation, the detailed mechanism of how SNORD15 affects expression of TRIM25 is not understood in detail. Also, the aspect of TRIM25 binding to snoRNAs is an interesting issue which should be addressed by future experiments.

Besides p53 modulation, other anti-apoptotic mechanisms by TRIM25 have been observed as well. The regulation of these pathways by TRIM25 will be briefly described in the following section.

#### 2.1.2. TRIM25-Controlled Pathways Acting Independently of p53

One of best-characterized examples supporting the strong pathological impact of TRIM25 is glioblastoma. In this type of cancer, high TRIM25 expression levels correlate with an unfavorable prognosis in patients with an aggressive subtype of glioma (WHO grade IV) [66]. Making use of diverse bioinformatics databases, including “Reactome Pathway” and the “Kyoto Encyclopedia of Genes and Genome” (KEGG) analyses, the authors of the study showed that the main biological functions of TRIM25 in glioma are attributed to its strong regulatory role on apoptosis and tumor immunity. Underlying pathways include NF-κB signaling and programmed cell death ligand (PD-L1)-related and macrophage-induced immune suppression pathways, which are probably the main reasons for the low success rates obtained with novel immune checkpoint interfering therapies in glioma patients [66]. In many cases, NF-κB is assumed as a key modulator of tumor cell survival. Consequently, silencing of TRIM25 inhibited the import of NF-κB into the nucleus and impaired expression of PDL-1, one of the downstream target genes of NF-κB. Data of this study clearly indicated that activation of the NF-κB-PD-L1 axis by TRIM25 critically contributes to the immune repressive microenvironment of therapy-resistant glioma. Mechanistically, TRIM25 promotes tumor necrosis factor α (TNFα)-induced NF-κB activation in various cell types, including macrophages, mainly through promoting the subcellular translocation of NF-κB by enhancing K63-linked TNF receptor associated factor 2 (TRAF2) ubiquitination via its RING domain [67]. Structurally, the K63 linked polyubiquitination of TRAF2 by TRIM25 provides a platform for bridging TRAF2 to transforming growth factor β activated kinase 1 (TAK1) or inhibitor of the NF-κB (IκB) kinase β (IKKβ) complex, resulting in the phosphorylation and subsequent proteasomal degradation of IκB [67]. Hence, TRAF2 itself has been described as a negative regulator of TNFα-induced apoptosis and necroptosis [68].

Apart from this, TRIM25 contributes to apoptosis resistance towards chemotherapeutic drugs by the inhibition of the phosphatase and tensin homolog (PTEN). PTEN is a tumor suppressor which exerts tumor-suppressive functions, mainly by impairing the phosphoinositide 3-kinase (PI3)/AKT/mTOR pathway relevant for cell growth and survival through dephosphorylating the 3- position of phosphoinositides [69]. Defects in PTEN have been described in many human cancers, including glioblastoma, prostate cancer, endometrial cancer, breast cancer, lung cancer, and melanoma, among others (for a review, see [70,71]), and are frequently due to gene mutations or, alternatively, to impaired posttranslational modifications, such as ubiquitination and phosphorylation [72]. In addition to the ubiquitin E3 ligase NEDD4-1 [73], PTEN is a target of TRIM25 which seems to impair PTEN activity through different modes of ubiquitination [74]. First, TRIM25 interacts with PTEN and mediates K63 linked polyubiquitination, which prevents the plasma translocation and phosphatase activity of PTEN, which results in the activation of AKT/mTOR signaling and increased resistance of non-small cell lung cancer cells (NSCLC) to cisplatin-induced apoptosis [75]. Conversely, the authors of the study observed that treatment of tumor cells with the antibacterial compound nitroxoline by counteracting TRIM25-mediated PTEN ubiquitination was able to restore cell death sensitivity of NSCLC to cisplatin [75]. Second, a study from HepG2 cells demonstrated that inhibition of the proteasome by MG132-blocked TRIM25-mediated reduction in PTEN protein content indicates a proteasome-dependent degradation of PTEN by TRIM25, which is most presumably initiated through K48-linked ubiquitination [76]. Furthermore, coimmunoprecipitation experiments revealed a direct physical interaction of TRIM25 with PTEN, and in line with this observation, ubiquitination by TRIM25 was markedly reduced upon TRIM25 silencing. These data suggest that TRIM25 is the ubiquitin ligase which mediates ubiquitination-dependent degradation of PTEN, which results in reduced sensitivity of tumor cells towards epirubicin. By addressing potential mechanisms underlying increased TRIM25 expression in NSCLC, a clinical study analyzing NSCLC tissues and adjacent normal tissues found elevated TRIM25 expression but low micro (miR) RNA-365 expression, mainly in tumor tissue [77]. Furthermore, the authors identified a functional miR365 binding site in the 3’UTR of human TRIM25 mRNA, and the relevance of this TRIM25-antagonizing miR was confirmed by the gain of function experiments demonstrating a strong reduction in TRIM25 expression concomitant with an increased expression of proapoptotic proteins in NSCLC cells. Collectively, the results from this study clearly indicate that targeting the TRIM25/PTEN/AKT/mTOR signaling pathway provides a promising strategy for resensitizing chemoresistant tumors to drug-induced apoptosis.

Another target of TRIM25-mediated proteolytic degradation to be mentioned is the alpha-2 Zinc-binding glycoprotein 1 (AZGP1), a soluble and secreted RNA-binding protein functionally involved in fat degeneration [78]. AZGP1 acts as a tumor suppressor mainly by inhibiting proliferation, migration, and invasion by negatively interfering with epithelial mesenchymal transition (EMT) induced by TGFβ signaling as well as PTEN/AKT/mTOR pathways [79,80]. Accordingly, low expression levels of AZGP1 were found in various cancers, including prostate and liver, and correlates with a worse patient prognosis [79,81]. For this reason, AZGP1 is discussed as a useful diagnostic marker in various cancers. A previous study on cholangiocarcinoma (CCA) identified TRIM25 as a negative regulator of AZGP1 and suggested that pathologically reduced AZGP1 may result from increased TRIM25 levels, since patients suffering from CCA were nearly deficient in AZGP1, but showed a high amount of TRIM25 [82]. Functionally, knockdown of TRIM25 in CCA cells induced apoptosis, as confirmed by Annexin/PI staining and increased cleavage of PARP, Caspase-3, and Caspase-9, and furthermore, it suppressed growth of tumor cells [82]. Mechanistically, TRIM25 interacts with AZGP1 through its PRY/SPRY domain, and thereby, targets its degradation by ubiquitination.

In contrast to mediating chemoresistance via promoting ubiquitination-triggered degradation of proteins with a suppressive activity on oncogenic signaling pathways, TRIM25 can stabilize epigenetic regulators of chemoresistance, as previously demonstrated for the enhancer of zeste 2 polycomb repressive complex 2 subunit (EZH2), a methyltransferase and part of the epigenetic polycomb repressive complex 2 (PRC2), which epigenetically dampens target genes through histone H3 tri-methylations [45]. Pathologically, an upregulation of EZH2, as observed in many different types of cancers, can essentially contribute to chemoresistance of CRC virtually through promoting cancer stem cell maintenance by modulating major intestinal stem cell renewal pathways such as Wnt/β-catenin and Hedgehog [83]. A recent study by Zhou and colleagues identified TRIM25-dependent stabilization of EZH2 as a novel epigenetic mechanism of oxaliplatin chemoresistance in CRC. Mechanistically, TRIM25 stabilizes EZH2 by impeding its proteasomal degradation by the E3 ligase TNF receptor-associated factor 6 (TRAF6) mainly by preventing TRAF6 binding to EZH2 via K63-linkage-dependent ubiquitination [45]. Although this pathway is not directly related to apoptosis resistance, it adds an important novel facet to the complex repertoire of mechanisms by which TRIM25 can promote tumor development and may, therefore, highlight the estimated benefits of TRIM25-targeting therapies for cancer treatment.

In a clear contrast to the so-far described scenarios in which TRIM25 has been commonly shown as an oncogenic protein which can essentially contribute to chemoresistance of many human cancers, data from nasopharyngeal carcinoma (NPC) clearly imply an opposite role of this particular TRIM member. By promoting polyubiquitination-dependent proteasomal degradation of KU80, a member of heterodimeric KU70/KU80 complex critically involved in the non-homologous end joining (NHEJ)-mediated DNA repair, TRIM25 enhances DNA damage concomitant with G2/M phase arrest and induces apoptosis of NPC [84]. However, this sensitizing mechanism is impaired in approximately 20% of NPC patients and accounts as a major reason for radioresistance. In these cases, TRIM25 protein stability is strongly reduced due to downregulation of the deubiquitinase USP44 as a result of hypermethylation of the USP44 gene promoter [84]. Thus, USP44 itself is unable to protect TRIM25 from proteasomal degradation by removing K48-linked polyubiquitination chains of TRIM25. The impaired USP44-TRIM25-KU80 axis is supposed to act as a major pathway of resistance of NPC and probably other human cancers towards ionizing radiation, and therefore, offers an attractive target for novel resensitizing tumor therapies.

While most of the tumor modulatory functions of TRIM25 arise from their ubiquitin- or ubiquitin-like posttranslational modifications of protein targets, including sumoylation, neddylation, and interferon-stimulated gene (ISG)ylation, plenty of data indicate that some of the pleiotropic effects by TRIM25 and various other TRIM members can additionally be attributed to a direct regulation of nucleic acids (for a review, see [85]). By employing gene set enrichment analysis, TRIM25 has, e.g., been identified as a key transcriptional regulator of metastasis-related breast cancer gene networks and knockdown of TRIM25 drastically impaired the expression of mainly those genes which are related to migration and invasion [86]. Data from our own laboratory revealed that in addition to controlling the transcriptome of tumor cells, TRIM25 affects post-transcriptional regulation of the pro-apoptotic caspase-2 (Figure 3) [87] and caspase-7 [88] via different mechanisms (Figure 4). Importantly, the modulatory effects on both caspases are due to either a direct or indirect binding to the corresponding mRNAs and functionally relevant for chemoresistance of colon carcinoma cells (Figure 3 and Figure 4). In this regard, TRIM25, together with some other RNA binding TRIM members, represents a subgroup of E3 ligases carrying RNA binding properties, as previously identified by high-throughput proteome analysis [89]. The fact that some E3 ligases, in addition to their originally described function in ubiquitin modification, share RNA binding capacities underscores the concept that protein regulatory processes can be tightly linked with the control of diverse RNA functions (for a review, see [90]). Using in vitro transcribed biotinylated mRNA encompassing the complete 5’UTR of caspase-2, we identified TRIM25 as a novel caspase-2 mRNA binding protein in human colorectal carcinoma cells and knockdown analysis confirmed that TRIM25 mainly interfered with the translation of caspase-2 mRNA without affecting caspase-2 mRNA contents (Figure 3). Importantly, TRIM25 binding to caspase-2 mRNA was markedly increased after CRC cells were exposed to doxorubicin, which corresponded to a reduction in caspase-2 translation, and thus, implicates a novel mechanism which is relevant for both intrinsic as well as acquired chemoresistance [87]. Functionally, results from TRIM25 knockdown experiments clearly indicate that TRIM25-mediated suppression of caspase-2 translation is relevant for resistance of CRC cells to genotoxic stress-induced apoptosis [87]. One of the best-described targets of caspase-2 is Mdm2, an RING-type E3 ligase which is described as the core negative regulator of p53, which keeps p53 contents at a low level in normal cells. However, in the case of CRC cells, the inhibition of Caspase-2 translation by TRIM25 was also observed in p53 deficient CRC cell lines, thus indicating that impaired Mdm2 cleavage is not relevant for the sensitizing effects by TRIM25 knockdown [87]. The identification of the molecular target of TRIM25, which is relevant for impaired translation of caspase-2, is an important issue which is currently investigated in our laboratory. In addition to intrinsic apoptosis, tumor-suppressive functions by caspase-2 are related to non-apoptotic functions such as genomic stability [91,92], autophagy [93], and control of the cell cycle checkpoint [94]. We have previously identified the DNA and RNA binding protein NONO (synonym: p54nrb) as a novel nuclear caspase-2 substrate and demonstrated that the loss/cleavage of NONO induced by genotoxic stimuli resulted in an attenuation of oncogenic gene expression and an increased cell death susceptibility of different tumor cell lines [95]. Functionally, NONO is engaged in almost every step of gene regulation, including pre-RNA processing and RNA transport, but also in the transcriptional regulation of oncogenic genes [96], including the anti-apoptotic gelsolin and the protease chatepsin-Z [95]. Thereby, NONO controls many key tumorigenic processes, including proliferation, migration, apoptosis, and DNA damage repair (for a review, see [97]). Accordingly, an overexpression of NONO was observed in different human cancers, including cervix adenocarcinoma, melanoma, and colon carcinoma, whereas depletion of NONO leads to an increase in tumor cell sensitivity towards drug-induced cell death and to an overall decreased tumorigenic potential [97]. Our finding indicates a novel mechanism of tumor-suppressive activities by caspase-2, and accordingly, inhibition of caspase-2-mediated NONO cleavage by TRIM25 may enrich the repertoire of transcriptional and posttranscriptional events [86] (Figure 3). Intriguingly, in addition to promoting tumorigenic functions by NONO indirectly via a suppression of caspase-2 translation, TRIM25 can directly enhance the splicing efficiency of NONO through K64-linked ubiquitination, which is, for example, functionally relevant for the splicing of arginine-methyltransferase-1 (PRMT1) and subsequent activation of the oncogenic c-MYC pathway in glioblastoma [98]. Therefore, targeting of TRIM25 via dual inhibition of NONO, on the one hand by restoring caspase-2-dependent NONO cleavage and on the other hand by impeding K63-linkage-dependent NONO ubiquitination, might turn out as a valid strategy for novel cancer therapy.

By searching for other apoptosis regulatory factors controlled by TRIM25, we identified the executioner caspase-7, which like caspase-2, was significantly upregulated in TRIM25-silenced CRC cells. However, in clear contrast to caspase-2, an inverse correlation between both genes was also found on the mRNA level and mainly related to TRIM25-mediated changes in caspase-7 mRNA stability [88]. Results from pull-down experiments showed that TRIM25 binds to the 3’UTR of caspase-7 mRNA via its PRY/SPRY domain and through interaction with the RNA-binding protein heterogeneous nuclear ribonucleoprotein H1 (hnRNPH1), which itself was able to mimic the negative effects on caspase-7 expression by TRIM25 (Figure 4). Interestingly, the interaction of TRIM25 with hnRNPH1 critically depends on RNA, implying that caspase-7 mRNA could serve as a bridging RNA, which allows a close interaction of TRIM25 and its target as a prerequisite for further ubiquitin-mediated posttranslational modification of hnRNPH1. Notably, hnRNPH1 levels were not affected by TRIM25 knockdown, ruling out that hnRNPH1 itself is a target of TRIM25-dependent proteasomal degradation [88]. Accordingly, data from co-IP experiments indicated that hnRNPH1 is a target of ubiquitin modification, although the specific K-linkage has not yet been determined. Functionally, silencing of TRIM25 caused a substantial increase in chemotherapy-evoked apoptosis, which is mainly attributed to an increase in caspase-7. Suppression of caspase-7 expression by TRIM25, probably executed through ubiquitination-dependent activation of the mRNA destabilizing factor hnRNPH1, implies a novel survival mechanisms of CRC cells towards genotoxic-induced apoptosis (Figure 4). Apart from acting as an executioner caspase, some non-apoptotic functions have been additionally ascribed for caspase 7, namely its role in signaling of the nucleotide-binding domain (NBD), leucine-rich repeat (LRR)-containing protein (NLR) family with a pyrin domain (NLRP)-3 inflammasome [99]. The modulation of these functions by TRIM25, in addition to apoptosis, has a critical impact on pyroptosis, as will be discussed later.

Mounting evidence from the literature indicates that several TRIM proteins, including TRIM11, TRIM19, and TRIM25, through mediating ubiquitination-dependent modifications, critically contribute to the degradation of misfolded proteins, and thus, play an emerging role in protein quality control, which is mainly localized in the endoplasmic reticulum (ER) [100,101]. Functionally, accumulation of misfolded proteins in response to ER stress is frequently linked to cancer and other diseases [102]. It is widely assumed that tumor cells have evolved specific signaling processes to restore ER homeostasis mainly through enhancing the capacity to eliminate misfolded proteins in response to unfavorable environmental conditions such as hypoxia, nutrient deprivation, or oxidative stress. Notably, aberrant activation of the unfolded protein response (UPR) together with ER-associated degradation (ERAD) seem highly relevant for the increased growth and survival of cancer cells and correlates with a bad prognosis in different types of cancers [103,104]. Another important pathway which essentially contributes to cellular resistance of tumor cells towards oxidative stress is the Kelch-like epichlorohydrin related protein-1 (Keap1)-nuclear respiratory factor (Nrf2) signaling pathway. Its activation results in the upregulation of antioxidant gene products, including heme oxygenase 1 (HO1) and NAD(P)H quinone oxidoreductase 1 (NQO1), mainly through binding to antioxidant response elements (AREs) in their gene promotors (Figure 5). Previous data from HCC identified TRIM25 as one of strongest induced genes in response to ER stress [104]. Functionally, TRIM25 bolstered tumor cell survival during ER stress mainly through directly targeting the Keap1 protein to ubiquitination-mediated proteasomal degradation, thereby promoting Nrf2 to the nucleus to activate the expression of Nrf2-responsive target genes of the anti-oxidative cell response (Figure 5) [104]. The fact that the Nrf2 pathway is one of prominent cell survival pathways that protect cancer cells from apoptosis inhibition of canonical TRIM25-Keap1-Nrf2 pathway implies another potential strategy for blocking cell survival pathways under chemotherapy.

### 2.2. Regulation of Ferroptosis by TRIM25

Ferroptosis, a recently identified novel mode of non-apoptotic iron-dependent cell death is mainly driven by excessive lipid peroxidation resulting from imbalances in cellular metabolism and redox homeostasis, finally leading to membrane damage and cell lysis. In a clear contrast to apoptosis, ferroptosis in most cases is pro-inflammatory, owing to the release of damage-associated molecular pattern molecules [105,106]. Pathologically, an escape from ferroptotic cell death is a characteristic feature of many diseases, including various types of cancer. Accordingly, increased resistance to this particular cell death modality correlates with resistance towards currently used tumor therapies, including radiotherapy, chemotherapy, immunotherapy, and targeted therapies [107,108,109,110]. Thus, ferroptosis is increasingly considered as a potential target of novel cancer therapy [111]. Like in HCC, TRIM25 propagates resistance of glioma to the chemotherapeutic drug temozolomide mainly through an inhibition of ferroptosis. Mechanistically, TRIM25 promotes the nuclear import of Nrf2 via ubiquitination-dependent degradation of Keap1 [112]. In a similar way, drug-induced expression of TRIM25 reduces ER stress by epoxy-eicosatrienoic acids (EETs) as products of eicosanoid metabolism, mainly through activating the Keap1/Nrf2/ARE axis, and thereby, it alleviates alveolar epithelial cell senescence [113]. Collectively, these studies confirm the role of TRIM25 as a critical mediator of ER homeostasis and as a promoter of resistance towards ferroptotic cell death.

In a search for endogenous peptides with potential anticancer effects in bladder cancer (BC), a previous study from Li et al. (2024) reported on the identification of cathepsin G-dependent protein 13 (CTSGDP-13) (Figure 5). This compound demonstrated potent anticancer effects in BC in vitro and in vivo mainly through an induction of ferroptosis [114]. Mechanistically, CTSGDP-13 interfered with the ferroptosis inhibitory activity of TRIM25, which itself is significantly upregulated in BC patients mainly through binding to the coiled-coil domain of TRIM25, and thereby, it impaired the binding of TRIM25 to the deubiquitinase USP7 (Figure 5). Consequently, the net increase in TRIM25 ubiquitination leads to its proteasomal degradation and restores ferroptosis. Furthermore, inhibition of the USP/TRIM25/Keap1 axis results in the impaired expression of the cysteine transporter and ferroptosis suppressor protein cystine transporter solute carrier family 7 member 11 (SLC7A11). Collectively, these data impressively illustrate the critical roles of TRIM25 in regulating ferroptosis.

In a clear contrast, another study on human pancreatic cancer cell lines, by employing an unbiased drug screening aiming on the identification of novel tumor cell selective ferroptosis-inducing compounds, identified TRIM25 inducers as a valid therapeutic approach to overcome ferroptosis resistance. Hereby, N6F11 was identified as a small molecule compound which by directly binding to the RING domain of TRIM25 triggered K48-linked ubiquitination and proteasomal degradation of glutathione peroxidase 4 (GPX4), a membrane lipid repair enzyme and master repressor of ferroptosis (Figure 6) [115]. Notably, degradation of GPX4 was restricted to cancer cells and did not occur in different types of immune cells. N6F11 may, therefore, represent an ideal candidate which via inducing ferroptotic cell death selectively in tumor cells could be therapeutically used for establishing a safer and more efficient strategy to boost ferroptosis-driven antitumor immunity [115]. In line with this report, a previous study by Yang and colleagues could demonstrate that TRIM25-mediated ubiquitination and subsequent proteasomal degradation of transcription factor SOX13 by the neuraminidase inhibitor zanamivir can reverse ferroptosis resistance in gastric cancer cells [116]. SOX13 induces the expression of splicing factor arginine/serine-rich 19 (SCAF1), and SCAF1-mediated assembly of respiratory chain super complexes, resulting in increased NADPH production in mitochondria, and thereby, it confers ferroptosis resistance. Accordingly, silencing of SCAF1 via induction of perturbations in the electron transfer chain in mitochondria affects proliferation and metabolism of different tumor cells and in vivo growth of some cancer types [116].

Together, these data indicate that depending on the tumor type, TRIM25 can exert opposing activities on ferroptosis, either positively via promoting proteasomal degradation of endogenous ferroptosis suppressors, or negatively by activation of USP/TRIM25/Keap1 axis and subsequent expression of ARE-driven antioxidant genes.

### 2.3. TRIM25 and Regulation of Autophagy

In addition to apoptosis and ferroptosis, autophagy represents another type of controlled cell death, where TRIM25 is a central part of a regulatory axis implied in chemotherapy resistance. Autophagy summarizes a cellular recycling pathway by which eukaryotic cells target cytoplasmic cargo for degradation to preserve intracellular homeostasis [117]. Targets of the autophagosomes, the major constituents of the autophagy machinery, include cytoplasmic protein aggregates, damaged organelles, but also invasive pathogens [118]. Principally autophagy primarily exerts protective and pro-survival (anti-apoptotic) roles in cells, but under certain conditions, autophagy can also lead to cell death, including iron-dependent ferroptosis [119,120]. Pathologically, dysregulated autophagy is implicated in various diseases, including cancer, where it mainly promotes cell survival. For this reason, autophagy inhibition is considered a promising approach for the treatment of most advanced cancers [121]. The regulation of autophagy is a common feature shared by several TRIM proteins via different mechanisms, including transcriptional modulation of autophagic gene expression, generation of a platform for recruitment of key autophagy regulators such as integrin-linked kinase 1 (ILK1), Beclin1, and others, or by affecting autophagosome formation by lipidation (for an up-to-date review, see [122]). In addition, some TRIMs by themselves can act as autophagy receptors for targeting substrates to autophagosome-mediated degradation [117]. In a previous study, Zheng and colleagues (2024) unraveled a novel mechanism by which TRIM25 promotes autophagy-triggered resistance of triple-negative breast cancer to taxanes [123]. Mechanistically, neddylation of TRIM25, a PTM akin to ubiquitination, by the E2 conjugating enzyme UBC12 increases the binding affinity of TRIM25 to its specific substrate, the transcription factor EB (TFEB), due to neddylation-dependent conformational changes in TRIM25 (Figure 7). Subsequently, TRIM25 via K63 linked ubiquitination promotes nuclear translocation of TFEB, thereby increasing transcription of some important autophagy-related genes, including ATG5, ATG7, and ATG12, which finally increases the resistance of triple-negative breast cancer to paclitaxel-induced cell death [123]. TFEB itself is assumed as a key regulator of autophagy (Figure 7), which is functionally linked to tumor development, and therefore, discussed as a putative target of novel tumor therapy [124,125]. It preferentially binds to the “GTCACGTGAC” motif found in the promoter regions of targeted genes and also referred to as coordinated lysosomal expression and regulation (CLEAR) element. Other studies have confirmed the potential benefit of neddylation inhibitors for tumor-sensitizing therapies to different chemotherapeutic agents [126,127,128]. Therefore, interference with single components of the autophagic UBC12/TRIM25/TFEB axis may offer a further pharmacological approach to restore the sensitivity of breast cancer and probably other cancer types to chemotherapy-induced cell death (Figure 7). However, data from additional in vivo studies are needed to substantiate these findings.

Another mode of controlled cell death path which has attracted increased attention during the last decade is pyroptosis. It describes a lytic cell death modality which is especially relevant in the context of inflammation and infection and which is characteristically mediated by inflammatory caspases, including caspase-1, -4, and -5, and is executed by members of the gasdermin family [129]. Pyroptosis is characteristically initiated by activation of an oligomeric protein complex containing a sensor protein, an adaptor protein, and the zymogen procaspase-1 summarized as inflammasome [130,131,132]. Principally, the assembly of inflammasomes is induced in response to a divergent range of pathogen-associated or danger-associated molecular patterns (PAMPs and DAMPs, respectively). The canonical inflammasome leads to an induction of caspase-1, which itself is required for the proteolytic maturation and release of interleukin (IL)-1β and IL-18, summarized as the activation phase of inflammasome signaling upon their transcriptional induction by NFκB during the priming phase of inflammasome activation [133]. Previous studies have highlighted the pivotal role of inappropriate activation of the inflammasome in the pathogenesis of prominent inflammatory diseases [132]. In the context of cancer, pyroptosis is known to exert mainly tumor-suppressive functions [134]. Although the impact of inflammasome signaling was mainly described in macrophages, the relevance of inflammasome-triggered processes in other cells, e.g., the intestinal epithelial cells, is increasingly recognized. Hereby, epithelial cell pyroptosis can occur downstream of the intrinsic apoptotic pathway, emphasizing the fact that different modes of regulated cell death can be interconnected at multiple levels. Meanwhile, several studies could highlight the regulatory role in inflammasome-mediated inflammation, namely by the Nod-like receptor pyrin-domain containing protein-3 (NLRP3) inflammasome by different TRIM proteins [135]. Importantly, some of the TRIM members, including TRIM16, TRIM28, TRIM33, and TRIM62, can promote activation of NLRP3 inflammasome signaling, e.g., by increasing ROS-NFκB signaling, which results in the elevated expression of single members of the inflammasome, namely NLRP3, pro-Caspase-1, and apoptosis-associated speck-like protein containing CARD (ASC) (for a review, see [136]). In contrast, some TRIMs, including TRIM25, can impair NLRP3 signaling indirectly through upregulation of antioxidant protein expression via induction of the canonical Keap1/Nrf2/ARE pathway described before, which negatively interferes with NLRP3 activation by downregulation of ROS. In addition, a direct inhibition of NLRP3 signaling can be achieved through TRIM-dependent K48-linked ubiquitination and degradation of NLRP3 as exemplary shown for TRIM31 [137] and TRIM65 [106]. In a similar manner, TRIM25 attenuates doxorubicin-induced pyroptosis in cardiomyocytes by decreasing the stability of NLRP-1 via promoting its ubiquitination [138]. However, to the best of our knowledge, a direct modulatory role of TRIM25 in pyroptosis was not described in cancer cells. This is less surprising, since pyroptosis represents regulated cell death modality mainly relevant for innate immunity and host defense by cytokine-triggered activation of immune regulatory cells, and thus, it is mainly implied in the pathogenesis of inflammatory diseases. Likewise, TRIM25 negatively interferes with the activation of the receptor interacting kinase 3 (RIPK3) in HEK293 cells, and thereby, inhibits TNFα-induced necroptosis by promoting K48-linked polyubiquitination-dependent degradation of RIPK3 (Figure 8) [139]. Necroptosis represents a further non-apoptotic mode of the immunogenic cell death pathway, which is induced by different death receptors, including Fas, TNFR1, TRAILR1, and TRAILR2, and the pathogen recognition receptor (PRR) [6]. In the case of TNF-induced necroptosis, RIPK3 and its family relative RIPK1 form the necrosome core which assembles to heterodimeric fibrils, a process which is driven by phosphorylation but only under conditions when caspase-8 is inhibited or depleted [140]. Phosphorylated RIP3 subsequently phosphorylates its downstream target the pseudokinase mixed lineage kinase domain-like (MLKL), thereby promoting oligomerization and translocation of MLKL from the cytoplasm to the plasma membrane, which results in pore formation and cell lysis (Figure 8) [141]. While the study highlighted TRIM25 as a potential target for RIPK3-dependent necrosis-related diseases, the possible impact of regulation of this particular cell death modality by TRIM25 on the therapy resistance of tumor cells has not been addressed so far and awaits further investigations.

## 3. Concluding Remarks

The role of TRIM25 in antiviral signaling pathways and innate immunity, particularly in the retinoic acid inducible gene 1 (RIG-1) pathway, is evidently demonstrated by many studies [26,27,28,29]. Despite these immunomodulatory actions, an increasing number of experimental data have documented an overexpression of this TRIM member in various human cancers. Thereby, in most cases, exaggerated TRIM25 expression correlates with a worse prognosis and poor outcome of patients. For that reason, TRIM25 is increasingly recognized as a valid tumor marker and promising target of novel antitumor therapy. Noteworthy, many of anticipated tumorigenic actions by TRIM25 and other oncogenic TRIM proteins rely on the prevention of apoptosis due to an inhibition of p53 stability and/or activity and therefore represents the predominant link to chemoresistance. Mechanistically, TRIM25 targets many proteins including p53 and accessory proteins for ubiquitin-mediated degradation. Along with modulating p53, TRIM25 can additionally interfere with other cell death pathways especially with pyroptosis, ferroptosis and autophagy. In line with its function as an RING-type of E3 ligase, therapeutic approaches which could specifically target the ligase activity of TRIM25 could be ideally used for (re)sensitizing tumor therapies. In this regard, it is worth mentioning that targeting of the proteasome has already been successfully implemented in the clinic by use of inhibitors of the 26S proteasome such as bortezomib, ixazomib, and carfilzomib [142]. Unfortunately, the therapeutic benefit of these approved inhibitors is restricted to the treatment of hematological malignancies, but could not be confirmed with solid tumors. Based on these rather humbling facts, exploring the benefits of strategies which interfere with the ubiquitin pathway upstream of the proteasome, namely with E3 ligases, has meanwhile turned into the focus of current research [142]. Specifically, targeting the RING subfamily of E3 ubiquitin ligases including TRIM25, which can confer specificity to the ubiquitin machinery, has shown promise against cancer in preclinical and clinical trials and is a widely recognized approach [142,143]. Notably, interfering with the activity of E3 ligases in addition to improving the effects of chemotherapies was shown to even increase the efficacy of current immunotherapy [144].

Practically, ongoing clinical trials with non-viral siRNA therapeutics to suppress the expression of specific E3 ligases by using different strategies, such as sequence modification, nanoparticle-mediated transport, or exosomal packaging, have achieved controlled delivery of siRNA therapies in living systems [145]. Apart from silencing approaches, small molecule inhibitors targeting specifically an individual E3 ligase, as previously realized by Hakin-1, a specific inhibitor of the E3 ligase Hakai, represents another promising therapeutic approach for interfering with cancer progression [146].

Another novel and elegant technology which makes use of the specific interaction between an E3 ligase and selected substrates to promote, for example, proteasomal degradation of a potential oncogenic substrate of a certain E3 ligase, termed PROTAC (proteolysis targeting chimera), makes advantage of synthetic bifunctional compounds bearing a chemical linker and two ligands. One of these ligands targets a specific E3 ligase and the other recognizes the protein of interest either via peptide binding, or via antibody-mediated recognition, thereby inducing exclusively the ubiquitination and subsequent degradation of selected target proteins [147,148]. Up to now, a high number of different PROTACs targeting different E3 ligases, including CRL4, von Hippel–Lindau (VHL), and Mdm2, have been tested in preclinical trials [149]. Whether some of the known oncogenic targets of TRIM25 described in this review may also be targeted by PROTAC is a challenging task which warrants further investigation.

## Figures and Tables

**Figure 1 cells-14-00065-f001:**
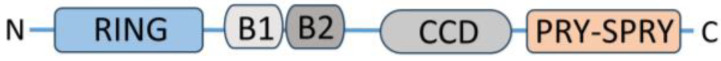
Schematic representation of the protein structure of TRIM25. B1/B2: B-box domain, CCD: coiled-coil domain.

**Figure 2 cells-14-00065-f002:**
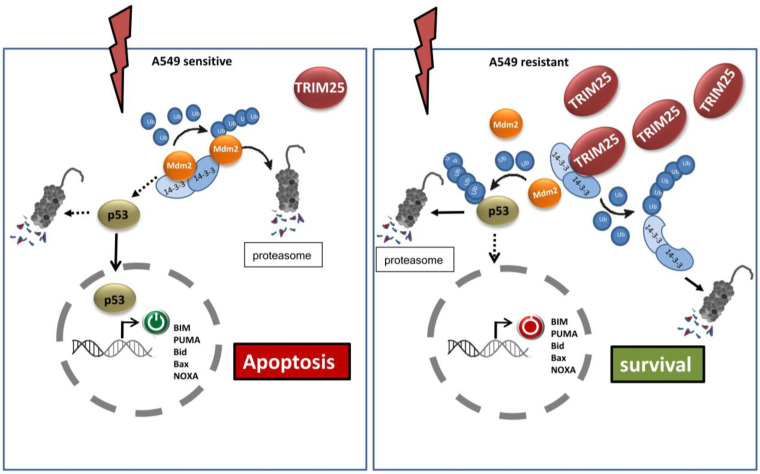
Proposed model of mechanisms underlying cisplatin resistance in non-small lung adenoma cells (A549/CDDP) (**right** panel) compared to drug sensitive parental A549 cells (**left** panel). Mechanistically, TRIM25-associated resistance to cis-CDDP is linked to an overexpression of TRIM25 but reduced levels of the scaffold protein 14-3-3σ, which itself is a target of TRIM25-dependent polyubiquitination and subsequent degradation by the proteasome. Consequently, the proteasomal degradation of the E3 ligase Mdm2 promoted by 14-3-3σ-dependent autoubiquitination (**left** panel) is impaired and leads to the facilitated proteolytic turnover of p53 and, finally, to a block in transcription of key apoptosis regulators by p53 and to resistance of cancer cells to drug-induced apoptosis. (**Left** panel) In contrast, low TRIM25 levels in A549 sensitive cells allow 14-3-3σ-accelerated autoubiquitination and degradation of Mdm-2, and thus, rescue p53 from Mdm2-triggered proteasomal degradation. Consequently, tumor cells are exposed to cisplatin, which induces nuclear import of p53, and subsequently, p53-dependent gene expression of pro-apoptotic factors. CDDP: cis-Diaminedichloroplatinum, Mdm2: murine double minute 2.

**Figure 3 cells-14-00065-f003:**
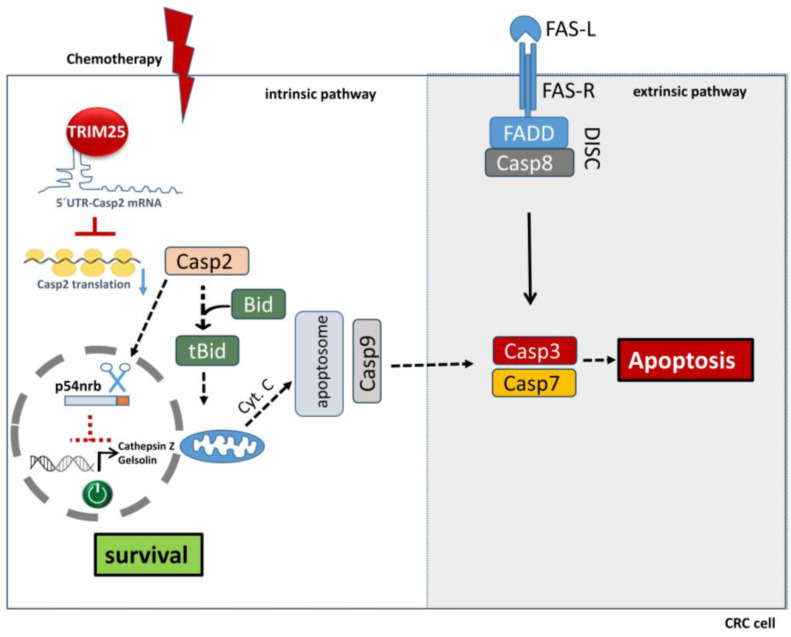
Postulated consequences of constitutive caspase-2 (Casp2) suppression by TRIM25 in colon carcinoma cells. (**Left** panel) Suppression of caspase-2 translation presumably results from constitutive TRIM25 binding to the 5’untranslated region (UTR) of caspase-2 mRNA and is functionally linked to impaired intrinsic apoptosis upon stimulation with chemotherapeutic agents, mainly due to the reduced cleavage of Bid and subsequent reduction in cytochrome c (cyt. C) release from mitochondria. In addition, impaired caspase-2 translation hinders the cleavage of the multifunctional protein p54nrb (NONO), and thereby, reinforces expression of tumorigenic genes, e.g., Cathepsin-Z and Gelsolin. Furthermore, this results in the impaired formation of apoptosome complexes and in the reduced activation of the executioner caspases-3 and -7 (Casp3, Casp7). (**Right** panel) As a further consequence, induction of the extrinsic and Fas ligand (FAS-L)- and caspase-8 (Casp8)-mediated apoptosis pathway by chemotherapeutic agents, e.g., doxorubicin, which converges on the activation of both executioner caspases, will be affected as well. Functionally, by increasing the thresholds for activation of both interconnected modes of apoptosis (intrinsic and extrinsic), TRIM25-mediated inhibition of caspase-2 translation may essentially contribute to the increased resistance of colon carcinoma cells towards chemotherapy-evoked apoptosis, and thus, constitute a so-far unrecognized survival program of tumor cells. Bid: BH3 interacting-domain death agonist, FADD: Fas-associated protein with death domain, FAS-R: FAS receptor, DISC: death-induced signaling complex.

**Figure 4 cells-14-00065-f004:**
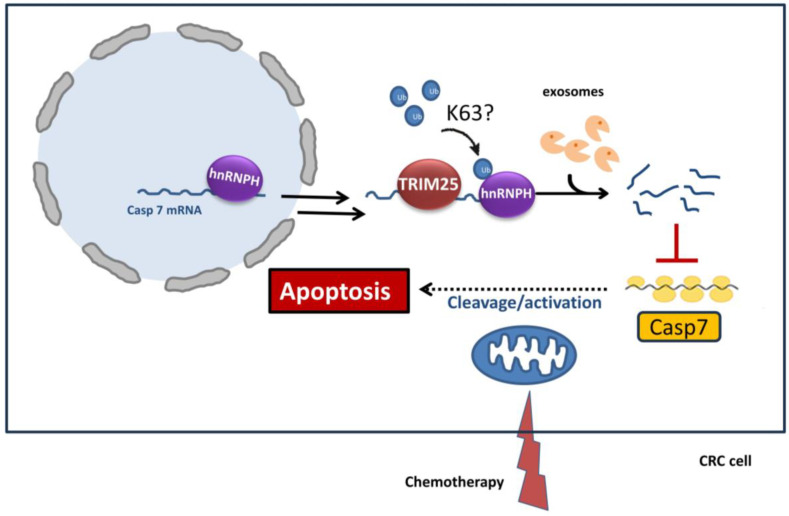
Inhibition of chemotherapy-induced apoptosis by TRIM25 via destabilization of caspase-7 (Casp7)-encoding mRNA. TRIM25, upon binding to the 3’UTR of caspase-7 mRNA and via binding and subsequent ubiquitination of the RNA binding protein hnRNPH1, suppresses the expression of caspase-7 mainly through destabilization of caspase-7 mRNA via exosome-triggered mRNA decay. The interaction of TRIM25 with hnRNAPH1 critically depends on RNA, thus indicating that caspse-7 mRNA serves as a “bridging RNA”, which facilitates physical interaction and ubiquitin-mediated posttranslational modification of hnRNPH1. Functionally, the reduction and impaired activation of caspase-7 (dotted arrow) critically contributes to increased resistance of CRC cells towards chemotherapeutic drug-induced intrinsic apoptosis. CRC: colon carcinoma cells; hnRNP: heterogeneous nuclear ribonucleoprotein; UTR: untranslated region.

**Figure 5 cells-14-00065-f005:**
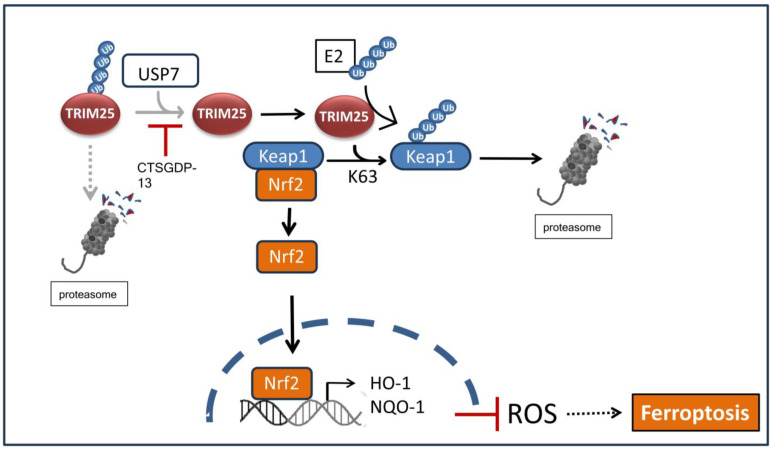
TRIM25 inhibits ferroptotic cell death by targeting Keap1 for proteasomal degradation. TRIM25 promotes the nuclear import of transcription factor Nrf2 via K63-triggered ubiquitination and subsequent proteasomal degradation of Keap1. Activation of Nrf2 leads to an induction of Nrf-responsive antioxidant gene expression as depicted, and consequently, to reduced ferroptosis due to an impaired ROS generation. Inhibition of the canonical TRIM25-Keap1-Nrf2 pathway in bladder cancer is achieved by the small peptide inhibitor CTSGDP-13 due to inhibition of TRIM25 with the deubiquitinase USP7, leading to an increase in ubiquitin-dependent degradation of TRIM25 (dotted arrow), and thus, to an increase in ferroptotic cell death. E2: ubiquitin conjugating enzyme; HO-1: heme oxygenase 1; Keap1: Kelch-like epichlorohydrin related protein-1; Nrf: nuclear respiratory factor; NQO-1: NAD(P)H quinone oxidoreductase 1; USP7: ubiquitin-specific protease.

**Figure 6 cells-14-00065-f006:**
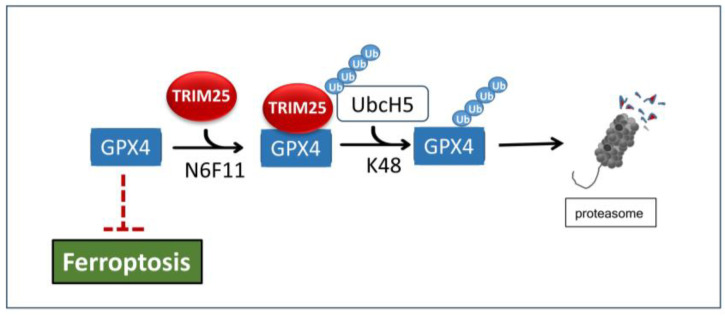
Induction of ferroptosis in pancreas cancer cells by the small-molecule compound N6F11 by increased recruitment of the ferroptosis inhibitor GPX4 to TRIM25. Consequently, GPX4 is targeted to TRIM25-triggered K48-linked ubiquitination and proteasomal degradation via UbcH5, thereby leading to an increase in ferroptotic cell death. GPX4: glutathione peroxidase-4.

**Figure 7 cells-14-00065-f007:**
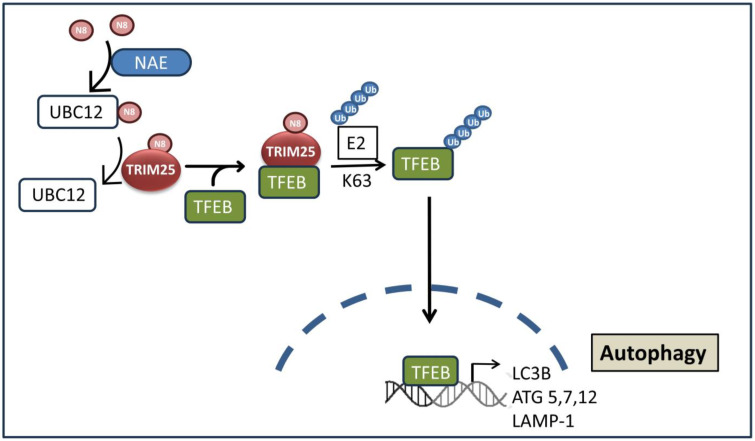
TRIM25 promotes autophagy-triggered resistance of triple-negative breast cancer cells to chemotherapy. Neddylation of TRIM25 by the E2 conjugating enzyme UBC12 increases the binding affinity of TRIM25 to TFEB due to neddylation-dependent conformational changes in TRIM25. Subsequently, TRIM25 promotes the nuclear import of TFEB via K63-linked ubiquitination, leading to the transcriptional induction of autophagy-related genes as depicted in the figure. Figure adopted from [122]. NAE: NEDD8 activating enzyme; TFEB: transcription factor EB; UBC12: ubiquitin conjugating enzyme E2M.

**Figure 8 cells-14-00065-f008:**
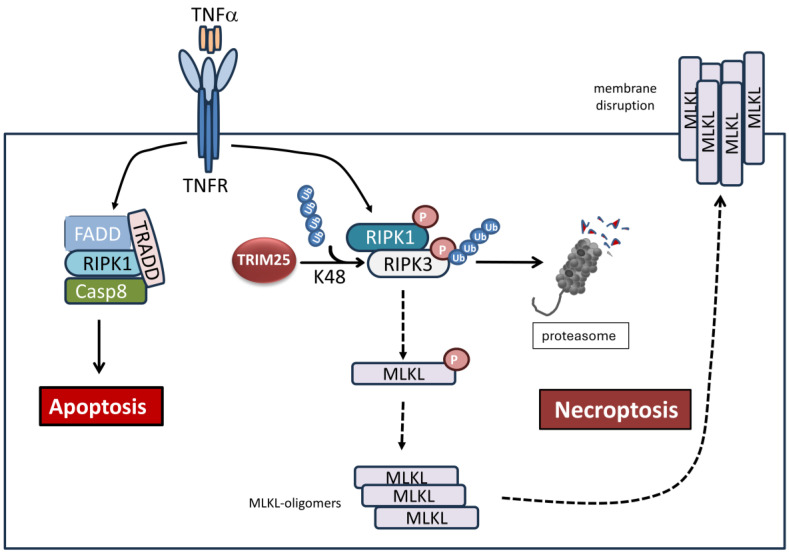
TRIM25 can negatively interfere with TNFα-induced necroptosis in HEK293 cells by promoting K48-linked ubiquitination and degradation of RIPK3. Proteasomal degradation of RIPK3 impairs the phosphorylation-dependent formation of the TNF-induced necrosome core complex. Consequently, phosphorylation of the pseudokinase MLKL by RIPK3, which is required for the formation and subsequent translocation of MKL oligomers to the plasma membrane is reduced. Thereby, pore formation and necroptotic cell lysis is impaired due to the TRIM25-dependent degradation of RIPK3. FADD: Fas-associated protein with death domain; MLKL: mixed lineage kinase domain-like, TNFR: tumor necrosis factor receptor, TRADD: TNFR1-associated death domain protein.

## Data Availability

Not applicable.

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
