# Peer review of "TRIM25: A Global Player of Cell Death Pathways and Promising Target of Tumor-Sensitizing Therapies"

_cells, 2025, doi:10.3390/cells14020065_

Round 1

Reviewer 1 Report

Comments and Suggestions for Authors

The authors have described the review report very well. This review will lay a strong foundation on the current knowledge of TRIM25 and will benefit reseachers. The report is well organised and can be accepted in the current form. 

Author Response

Reviewer 1

The authors have described the review report very well. This review will lay a strong foundation on the current knowledge of TRIM25 and will benefit researchers. The report is well organized and can be accepted in the current form. 

Response: We thank the reviewer for his positive comments and are happy about his appreciation.

Reviewer 2 Report

Comments and Suggestions for Authors

. Given the current state of research in apoptosis and drug resistance field, the manuscript lacks critical depth in its review of current challenges and future directions in the field.

having a graphical abstract can illustrate better the massive information of pathways and prospects suggested in the papr.

Author Response

Comment 1: Given the current state of research in apoptosis and drug resistance field, the manuscript lacks critical depth in its review of current challenges and future directions in the field.

Response 1: We thank the reviewer for his critical comments. We want to emphasize on the fact that is very difficult to find an ideal compromise between preparing a most comprehensive review on the one hand but at the same time to limit the information flood which is inevitable when trying to give a state of the art review. To best of our knowledge our review is the first review which comprehensively summarizes the current knowledge of TRIM25 actions on diverse cell death pathways and its particular roles in tumorigenesis. A particular focus on how to translate this knowledge in clinical directions and applications could ideally be content of a separate follow-up article.

Comment 2: having a graphical abstract can illustrate better the massive information of pathways and prospects suggested in the paper.

Response 2:

As suggested by the reviewer, we have now added a graphical abstract to summarize the different cell death pathways controlled by TRIM25.

Reviewer 3 Report

Comments and Suggestions for Authors

In their manuscript Eberhardt et al. reviewed the function of TRIM25 globally, and more specifically in the context of cell death, tumorigenesis and resistance to chemotherapies. This is a very exhaustive and well-illustrated review that addresses most, if not all, the functions of TRIM25. It will be a valuable tool for researchers studying this protein.

A few suggestions:

-          I think a figure with the structure of TRIM25 and its different domains could be beneficial for the readers.

-          In figure 2 the authors describe the interaction of TRIM25 with capase 2. In the text (lines 469-493) they described the link between NONO, TRIM25 and caspase 2. The authors could contemplate the possibility of including NONO in fig 2.

-          Lines 198-200 : In most cases 14-3-3 by accelerating the proteolytic turnover of Mdm2 through enhancing auto-ubiquitination of Mdm2 which leads to the stabilization of p53 and subsequently, to activation of p53 related tumor suppressive functions.

This sentence is not clear. Perhaps remove “which”?

 Some sentences are very long and difficult to follow.

For instance, line 212-217:

In this regard it is worth mentioning that in the context of breast cancer, the use of TRIM25 targeting DNA-based “chimeric siRNA” in which a part of the sequence is substituted with DNA are supposed to have less off-target effects or immune responses in mammalian cells com-pared to conventional siRNA duplexes, efficiently suppressed proliferation and cell cycle progression of MCF-7 cells, as well as in vivo growth of MCF-7-derived tumors in nude mice.

Small detail: in several places the references for reviews are located after the brackets

for instance lines 36: “(for review see) [4, 5]” except on line 153  “(for a review see [35])”. It looks better with the references inside the brackets.

Author Response

Reviewer 3

In their manuscript Eberhardt et al. reviewed the function of TRIM25 globally, and more specifically in the context of cell death, tumorigenesis and resistance to chemotherapies. This is a very exhaustive and well-illustrated review that addresses most, if not all, the functions of TRIM25. It will be a valuable tool for researchers studying this protein.

A few suggestions:

-          I think a figure with the structure of TRIM25 and its different domains could be beneficial for the readers.

-          In figure 2 the authors describe the interaction of TRIM25 with caspase 2. In the text (lines 469-493) they described the link between NONO, TRIM25 and caspase 2. The authors could contemplate the possibility of including NONO in fig 2.

-          Lines 198-200: In most cases 14-3-3 by accelerating the proteolytic turnover of Mdm2 through enhancing auto-ubiquitination of Mdm2 which leads to the stabilization of p53 and subsequently, to activation of p53 related tumor suppressive functions.

This sentence is not clear. Perhaps remove “which”?

Response:

We thank the reviewer for his most helpful comments.

Response 1: According to his suggestion we have now added a new Figure 1 depicting the structure of TRIM25 and his different domains.

Response 2: We thank the reviewer for this important issue and according to his suggestion, we have now changed Figure 2 and accordingly the text of the associated figure legend and included NONO as potential target of caspase-2 and its putative implication in modulation of tumor gene expression.

Response 3:

We thank the reviewer for his attention. As suggested, we have corrected this sentence and removed the corresponding word.

Round 2

Reviewer 2 Report

Comments and Suggestions for Authors

I still couldnt recognise which figure is for graphical abstract

the figure numbers are not correct and misleading